# Does Endometriosis Influence the Embryo Quality and/or Development? Insights from a Large Retrospective Matched Cohort Study

**DOI:** 10.3390/diagnostics10020083

**Published:** 2020-02-03

**Authors:** Ana M. Sanchez, Luca Pagliardini, Greta C. Cermisoni, Laura Privitera, Sofia Makieva, Alessandra Alteri, Laura Corti, Elisa Rabellotti, Massimo Candiani, Paola Viganò

**Affiliations:** 1Reproductive Sciences Laboratory, Division of Genetics and Cell Biology, IRCCS San Raffaele Scientific Institute, 20132 Milano, Italy; asg.alcala@gmail.com (A.M.S.); makieva.sofia@hsr.it (S.M.); vigano.paola@hsr.it (P.V.); 2IRCCS San Raffaele Scientific Institute, Obstetrics and Gynecology Unit, 20132 Milano, Italy; cermisoni.greta@hsr.it (G.C.C.); privitera.laura@hsr.it (L.P.); alteri.alessandra@hsr.it (A.A.); corti.laura@hsr.it (L.C.); rabellotti.elisa@hsr.it (E.R.); candiani.massimo@hsr.it (M.C.)

**Keywords:** embryo quality, endometriosis, blastulation rate, ongoing pregnancy

## Abstract

In vitro fertilization can be an effective tool to manage the endometriosis-associated infertility, which accounts for 10% of the strategy indications. Nevertheless, a negative effect of endometriosis on IVF outcomes has been suggested. The aim of this study was to evaluate the potential effect of endometriosis in the development of embryos at cleavege stage in assisted reproduction treatment cycles. A total of 429 cycles from women previously operated for moderate/severe endometriosis were compared with 851 cycles from non-affected women. Patients were matched by age, number of oocyte retrieved and study period. A total of 3818 embryos in cleavage stage have been analyzed retrospectively. Overall, no difference was found between women with and without endometriosis regarding the number of cleavage stage embryos obtained as well as the percentage of good/fair quality embryos. Excluding cycles in which no transfers were performed or where embryos were frozen in day three, no difference was observed for blastulation rate or the percentage of good/fair blastocysts obtained. Despite similar fertilization rate and number/quality of embryos, a reduction in ongoing pregnancy rate was observed in patients affected, possibly due to an altered endometrial receptivity or to the limited value of the conventional morphological evaluation of the embryo.

## 1. Introduction

Endometriosis affects from 10% to 15% of reproductive aged women and around 30% of women suffering from infertility, which is up to 10-fold more frequent than in the general population (0.5–5%) [1,2]. Mechanisms that have been postulated to explain the low fecundity of women with endometriosis include altered folliculogenesis, reduced quality and cytoplasmic mitochondrial content of oocytes, oocyte/embryo exposure to a hostile inflammatory environment (macrophages, cytokines and vasoactive substances in the peritoneal fluid), anatomical dysfunctions of the tubes and/or ovary and reduced endometrial receptivity [3].

Assisted Reproduction Technology (ART) can be an effective tool to manage the endometriosis-associated infertility, which indeed accounts for 10% of the strategy indications. Nevertheless, a negative effect of endometriosis on ART outcomes has been suggested [4,5,6], albeit not consistently [7,8,9]. Both oocyte/embryo number and quality have been claimed to be affected by the disease [3]. Lower implantation rates have been as well postulated. However, the reasons to explain the suboptimal performance of ART in endometriosis patients are still poorly understood and can only be hypothesized.

Data from meta-analyses are only partially informative in this regard. The meta-analysis from Barnhart et al. [4] including data from 22 studies, found a reduction in fertilization and implantation rates in women with endometriosis when compared with non-affected women or with women that underwent ART for tubal-factor infertility only [4]. Harb and colleagues, including 27 studies, reported a reduction in clinical pregnancy rate in women with stage III/IV endometriosis compared to controls, but not a reduction in live births [10]. No differences in reproductive outcomes were found by Barbosa and colleagues between women with and without endometriosis. Only the number of oocytes at the time of retrieval was found to be lower in women affected [11].

Unfortunately, few studies have considered the consequence of endometriosis on the embryological outcomes. This aspect is important considering that in recent years the reproductive medicine laboratories are trying to optimize embryo transfer strategies, e.g., by transferring the embryo later in development instead of transfer at an early stage. Additionally, single blastocyst transfer has been preferred to a simultaneous transfer of multiple early stage embryos.

In association with the importance of the embryonic developmental stage for an optimal uterine transfer, it is critical to elucidate factors that can threaten embryonic competence to progress in a healthy pregnancy. In this context, Freis and colleagues have recently reported that the relative morphokinetic profiles of embryos from patients with endometriosis are altered, indicating a negative impact of the disease independently from the stage on the embryo quality [12]. Herein, we have scrutinized the plausible negative impact of endometriosis on embryonic parameters in a retrospective non-interventional analysis of ART cycles in our center.

The aim of the study was to investigate whether endometriosis affects embryo development and/or quality. The primary outcome of the study was the quality of cleavage stage (day 3) embryo in terms of number of cells, cell fragmentation and symmetry. Secondary outcome measures were (1) fertilization rate, (2) number of good/fair embryos at cleavage stage, (3) blastulation rate (defined as percentage of total blastocyst obtained per number of fertilized eggs, excluding cycles in which embryos were transferred or frozen in day 3, (4) good/fair blastocyst formation rate, and (5) ongoing pregnancy rate (defined when the pregnancy had completed ≥20 weeks of gestation per transfer).

## 2. Results

The baseline characteristics of the cycles for the two study groups are presented in Table 1: the maternal age, Body Mass Index (BMI), antral follicle count and Anti-Müllerian hormone (AMH) levels were significantly different between the two groups.

Age was used as a variable to match cases and controls and a significant difference was observed between the two groups, that was however limited as an absolute value and for which the subsequent results were adjusted. The semen characteristics of the endometriosis and non-endometriosis groups are shown in Table 1. Sperm concentration and motility were significantly different between the groups. No differences were found in the levels of estrogen at the time of hCG administration and in the number of oocytes retrieved per 1000U of FSH (2.8 ± 4.1 controls vs 2.2 ± 2.6 endometriosis patients, *p* = 0.09).

The number of oocytes retrieved in both groups was similar (5.8 ± 4.4 non-endometriosis vs 5.8 ± 4.6 endometriosis patients, *p* = 0.9). We did not find any statistically significant differences in the percentage of MII oocytes (75% (50–100%) controls vs. 71% (50–100%) endometriosis patients, *p* = 0.25) and in fertilization rate (75% (50–100%) controls vs. 75% (50–100%) endometriosis patients, adjusted *p* = 0.85) (Table 2).

Overall, we did not find any difference regarding the number of cleavage stage embryos obtained (3.5 ± 2.6 non-endometriosis vs. 3.7 ± 2.6 endometriosis patients, adjusted *p* = 0.77) and the mean number of blastomers of the embryos (7.0 ± 1.4 controls vs 6.9 ± 1.5 endometriosis patients, adjusted *p* = 0.42) between endometriosis women and controls. In addition, the percentage of good/fair quality embryos was similar (56% (25–100%) controls vs. 50% (17–88%) endometriosis patients, adjusted *p* = 0.36). Excluding cycles whereby embryos were transferred or frozen in day 3, no difference was found in blastulation rate between the two groups (50% (25–67%) controls vs. 50% (25–67%) endometriosis patients, adjusted *p* = 0.22). Finally, we calculated the percentage of good/fair blastocysts obtained and we did not find any difference (50% (33–80%) controls vs. 50% (25–75%) endometriosis patients, adjusted *p* = 0.88) (Table 2). No differences in terms of the number of cancelled cycles and/or freeze-all cycles were found between both groups (Appendix A). 

Despite similar fertilization rate and number/quality of embryos obtained, we found a reduction in ongoing pregnancy after adjusting for the number of transferred embryos and the day of transfer (cleavage or blastocyst stage) (24.2% in controls vs. 17.8% in endometriosis group, adjusted OR = 0.62; 95% CI 0.40–0.94, *p* = 0.02) (Table 3).

A similar reduction was observed when considering transfers at day 3 or day 5, separately. Similar results were observed in terms on ongoing pregnancy rate considering frozen embryo transfers (Appendix A).

## 3. Discussion

This is, at least to the best of our knowledge, the largest study that analyzed the potentially deleterious effect of endometriosis on the in vitro development of embryos in ART cycles. We were unable to demonstrate an impact of endometriosis on day three embryo quality and developmental potential. The fertilization rates and percentage of good/fair quality embryos from endometriosis patients and controls were also similar. Moreover, in patients who did not perform a fresh transfer and/or freeze embryos at day three, we did not find any statistical difference in blastulation rate and/or the percentage of good/fair quality blastocyst obtained. Therefore, in the light of the results obtained, women with endometriosis may as well opt for the blastocyst culture in the presence of good quality embryo at day three in order to improve the reproductive outcomes after ART [13]. 

A limited number of studies have been published in relation to the in vitro development of embryos obtained from women with endometriosis. Coccia and colleagues published one of the first studies in 2011. In contrast to our results, the total number of embryos obtained in their ART cycles was significantly different between the endometriosis and the control group represented by women with tubal factor infertility. This discrepancy may be explained by methodological differences: firstly, they did not match for age, number of oocytes retrieved and/or study period. Secondly, only 3 oocytes were used for conventional IVF (under Italian law 40/2004 for ART) [5]. Finally, in endometriosis patients they notably observed a decrease in the number of oocytes retrieved, which could have impacted the number of embryos obtained. The reduction in the number of oocytes retrieved demonstrated in several studies may be ascribed to the detrimental effect of previous surgical treatments rather than to the disease itself. It is for this reason that we have decided to match our population for this parameter in order to avoid this bias. More recently, two studies have been performed using the time-lapse technology for the assessment of embryo morphokinetics in endometriosis patients [12,14]. In the study by Demirel and colleagues, the endometriosis population was constituted only by patients with a diagnosis of endometrioma. Specifically, the authors compared embryos derived from oocytes collected from an ovary affected by an endometrioma to embryos developed from oocytes from the contralateral healthy ovary failing to find differences in terms of morphokinetic parameters [14]. In contrast, Freis and colleagues compared embryo morphokinetics between women with and without endometriosis (tubal factor) showing a poorer relative kinetics in embryos from affected patients [12]. 

The group of Song and colleagues have demonstrated that the number of mature follicles and good embryos, and fertilization and blastulation rates were reduced in women with endometriosis compared with women with a male factor indication [15]. In line with our results, Benaglia and colleagues found that, in women with bilateral endometriomas, despite the lower number of oocytes retrieved, no differences could be observed in terms of fertilization rate, number of embryos obtained and rate of top-quality embryos per oocyte used compared to controls without the disease [16]. Finally, in a recent work of Muteshi and colleagues they demonstrated that endometriosis may affect embryo development due to a reduction in the percentage of women with endometriosis that reach blastocyst transfer compare with women with unexplain infertility [17]. Therefore, overall, data from the literature addressing the embryological parameters in ART cycles of women with endometriosis are very controversial. 

Unfortunately, some of these previous studies are characterized by important limitations: firstly, the small sample size resulting in lack of statistical power and secondly, the lack of matching for age or number of collected oocytes or of corrections for confounders. These limitations of others have been addressed in the present study and represent the main strength of our work. Based on our results, at present, there are no strong evidence to set up a different culture or transfer strategy or to change the conventional embryological practice in ART cycles performed in patients with endometriosis.

In terms of IVF clinical outcomes, despite a similar number of transferred embryos and no differences in the day of transfer between women with and without the disease, we found a reduction in the ongoing pregnancy rates in the affected women. Several meta-analyses describing the effect of endometriosis on IVF outcomes have been published, again with contradictory results [4,6,10,11]. The inclusion of studies with very heterogeneous populations both for endometriosis and control group represents the main problem of these meta-analyses. The presence of side causes of infertility other than endometriosis are also rarely considered in the meta-analyses [18]. It should be considered that our endometriosis population consists of all operated women for a stage III/IV disease and that we have corrected for confounders potentially affecting fertility such as BMI and sperm motility/concentration.

In terms of ongoing pregnancy rate we found a similarly reduction of ongoing pregnancy rate in the endometriosis group compared to the control one in the freeze all cycles, consequently we cannot conclude that endometriosis women have better ART outcome after freeze all cycle. This data is supported by the recent study of Roque and colleagues [19], that reported that even if initial studies showed that the freeze-all strategy could be beneficial for certain groups of infertility including endometriosis patients [20], there is still lack of evidence to support its routine use not only for indications as endometriosis but also for implantation failure. In conclusion the data published until now is very controversial about better ART outcomes after freeze all cycles in endometriosis patients [19].

Two hypotheses can be put forward to explain our overall findings. The first is that, given the lack of differences in terms of embryo development/quality and blastulation rate in the endometriosis group, one might wonder whether an altered endometrial receptivity may explain the reduced chance of an ongoing pregnancy in affected women. Inflammatory-related changes in gene expression and/or a progesterone resistance in the endometrium of women with endometriosis might have a role [21,22]. This again represents a critical point. Some endometrial receptivity defects have been detected in these women [23]. However, data from clinical IVF egg donation program do not support this idea [24].

The second explanation is that the conventional morphological evaluation of the embryos, even at day 3 of development, may scarcely predict the embryo competence in these patients. Embryo grade has some value in predicting implantation [25] but, certainly, the embryo selection based on morphologic criteria could be imprecise [26] and may be even more imprecise in women with endometriosis. Indeed, oocytes have been demonstrated to more likely fail in vitro maturation and to have lower cytoplasmic mitochondrial content in women affected compared to women with other causes of infertility [3]. The time-lapse technology might be of value in this context.

The main limitation of our study is its retrospective design; however, for the calculation of the main outcome, no clinical decision/intervention has been done between the time of fertilization and day three of culture. The majority of the women from control group did not undergo laparoscopy prior to ART, therefore we cannot totally exclude the possibility that we incorrectly selected some affected women. This possibility could have influenced the study power to detect differences between the two groups. Similarly to the study of Barbosa and colleages, we have considered to include in the control group all the other women (without the diagnosis of endometriosis), because applying any other selection criterion would be arbitrary and might introduce biases [11]. Finally, the heterogeneity of the control group in terms of cause of infertility represents another bias that should be considered. Indeed, the different prevalence of causes underlying infertility could significantly impact on the quality of the embryos analyzed. This problem may be one of the reasons for the disagreements in terms of ART outcomes in the different studies already published and might have impacted the outcomes related to the blastulation rate and pregnancies. Similarly, although all cases were post-surgical, parameters such as moderate/severe disease, the different intervals from surgical management of endometriosis and ART treatment may have affected the results. In fact, based on a recent study of AlKudmani and collaborators [27], significant higher IVF ongoing pregnancy rates were observed in women with endometriosis after 6–25 months from the surgery compared with women with endometriosis undergoing IVF after 25 months from surgery [27]. In addition, another limitation of the study is the lack of information about the location of endometriosis lesions (superficial endometriosis, deep infiltrating endometriosis, endometriomas). At this regard, in a recent study of Ashrafi and colleagues [28], they demonstrated that the presence of deep infiltration endometriosis in the presence or not of endometrioma was associated with an 80% decrease in the probability of live births in comparison with that of a control group, but no information concerning the performance in the laboratory has been published. 

In conclusion, we report that endometriosis does not compromise fertilization rate, the quality of cleavage stage embryos, of the blastocysts and blastulation rates. Thus, this study does not support the need to tailor embryo transfer strategy to the incidence of endometriosis. On the other hand, we found a reduction in ongoing pregnancy in patients affected, but an explanation for this observation warrants certainty. These results can help in understanding the mechanisms by which endometriosis impacts fertility.

## 4. Materials and Methods

### 4.1. Patients

This is a single center retrospective study matched cohort study, non-interventional, including patients who underwent ART cycles at the San Raffaele Hospital (Milan, Italy) from 2013 to 2017. 

We performed a matched cohort study of the variables believed to be confounding (in order to avoid confounding) in our study design and to ensure an equal distribution among affected and non-affected. 

Endometriosis was laparoscopically diagnosed in all the patients before ART treatment and classified according to the American Society for Reproductive Medicine ASRM criteria into stage III-IV (moderate/severe) [29], all women received a complete surgical treatment for endometriosis and all the lesions were removed. A total of 309 women were included in the study with a total of 429 ART cycles performed. 

The control group were patients without a laparoscopic diagnosis or a history of endometriosis and did not have any ultrasonographic evidence of endometriotic ovarian cyst at the time of the cycle, including patients with tubal factor (female infertility caused by diseases, obstructions, damage, scarring, congenital malformations or other factors that impede the descent of a fertilized or unfertilized ovum into the uterus through the Fallopian tubes), male factor (patients that underwent at least two consecutive semen analyses, both showing below-standard values for normal semen parameters according to the World Health Organization (WHO) criteria), poor ovarian reserve (Bologna criteria definition [30]), idiopatic or patients that underwent PGT-M cycles for monogenic diseases (diseases that not affect the fertility of the couple, excluding genetic causes of male infertility) (Appendix A). Controls were matched to cases on a ratio 2:1 by age (± 1 year), number of oocytes retrieved (± 1 oocyte) and study period (± 4 months). A total of 851 cycles from control women were included in the analysis (from 766 patients). A total of 3818 cleavage stage embryos (day 3) have been analyzed.

Data collection followed the principles defined in the Declaration of Helsinki; all women undergoing ART in San Raffaele Hospital routinely provide informed consent for their clinical data and anonymized records to be used for research purposes in general. Women who denied this consent were excluded. Local Institutional Review Board approval (ID: BC-GINEOS, date of approval: 09/02/2012, San Raffaele Hospital Ethics Committee) for the use of clinical data for research studies was obtained.

### 4.2. Controlled Ovarian Stimulation, In Vitro Fertilization, Embryo Culture and Grading

All patients were treated with a GnRH antagonist protocol as previously descrived [31]. Oocyte collection was performed 36 h after triggering of ovulation. In both groups, in order to avoid biases in the evaluation of the number of MII oocytes and in the fertilization rate, only ICSI cycles were considered for analysis (that represents the 91% of the study group before matching). ICSI cycles were performed as previously descrived [32]. Sixteen-eighteen hours after ICSI, oocytes were checked for fertilization and transfer to 10% of Serum substitute supplement (SSS, Irvine, CA, USA)-supplemented Cleavage medium (REF ART-1026, Sage In-Vitro Fertilization, inc. Trumbull, CT, USA) under oil. Embryos were checked at 68 ± 1 after ICSI, and an embryo evaluation was performed according to the Istanbul consensus [33]. Briefly, the embryo quality was calculated in terms of number of blastomers, cell fragmentation and symmetry. A good/fair-quality embryo was considered that with ≥7 cells on day 3, with a fragmentation rate lower than 10% and a stage-specific cell size for majority of cells [33]. 

For evaluation of blastocyst data [cultured into 10% of Serum substitute supplement (SSS, Irvine, CA, USA)-supplemented Blastocyst medium (REF ART-1029, Sage In-Vitro Fertilization, inc. Trumbull, CT, USA)], only patients who did not perform transfer or freezing of day 3 embryos were included, hence patients who underwent prolonged culture of the whole cohort of embryos formed.

Blastocyst evaluation was performed according to the Istanbul Consensus (116 ± 2 h post insemination) [33] as previously described [31]. Based on the rating, a good/fair blastocyst was defined as expanded or hatched blastocyst with both an inner cell mass and multicellular trophectoderm scored good or fair or at least one of them scored as good or fair. Blastocysts were never frozen before the expanded stage.

All embryo transfers were scheduled in day 3 (at cleavage stage). The decision to transfer in day 3 or delay the embryo culture to day 5 (blastocyst stage) depends on the evaluation of the quality and the number of cleave stage embryos at day 3 and woman’s age. Criteria for blastocyst culture were the presence at day 3 of 4 or more embryos and at least 2 good/fair embryos in women younger than 38 years old and three good/fair embryos for women aged 38 years and older [34].

In our laboratory, an embryo quality control is performed with a biannual frequency, together with other subjective evaluations (i.e., oocyte quality control, preimplantation diagnosis biopsy control) among the embryologists as previously described [31,35]. In any case, for this specific study, we choose to enroll in the study only cycles in which the quality of the embryos was evaluated every day by the same embryologist (P.V.) who is assigned to this specific task most of the time. To prove the reliability of the evaluations, we have measured the ability of conventional morphological analysis of the embryo to predict ART outcomes. An odds ratio of 5.7 (95% confidence interval 1.7–19.5, *p* = 0.006) for ongoing pregnancy was found after a single embryo transfer of day 3 good/fair embryos compared to poor quality embryos.

### 4.3. Statistical Methods

All continuous variables are presented as mean ± standard deviation (SD) or median with interquartile range (IQR). Normality of variables distributions were checked using the Kolmogorov–Smirnoff test. Student’s *t*-test, Mann–Whitney U test and chi-squared test were used as appropriate. Multivariate analysis was conducted using the generalized estimating equation (GEE) approach, thus making it possible to use multiple cycles of the same patient and at the same time allow the analysis of variables with non-normal distribution. Age, BMI, concentration and motility of sperm and percentage of mature oocytes were used as predictors in the GEE model in order to obtain adjusted estimation for the differences between cases and controls. The ongoing pregnancy rate was adjusted for day of transfer and also for the number of transferred embryos and the adjusted estimate was reported as odds ratio (OR) with 95% confidence interval (95% CI). Differences were considered statistically significant if *p* < 0.05. Data were analyzed using the SPSS software 17.0 (SPSS, Inc., Chicago, IL, USA).

## Figures and Tables

**Table 1 diagnostics-10-00083-t001:** Basal characteristics of the analyzed cycles.

Parameters	Controls *n* = 851	Endometriosis *n* = 429	*p*-Value
Age (years)	37.5 ± 3.6	36.9 ± 3.6	0.003
BMI (kg/m^2^)	22.4 ± 3.6	21.6 ± 2.9	0.004
Antral follicle count	8.0 ± 4.7	6.3 ± 3.2	<0.001
AMH (ng/mL)	1.9 ± 2.5	1.5 ± 1.9	<0.001
Total dose FSH administered (IU)	3112 ± 1600	3419 ± 1641	0.003
E2 at the time of hCG administration (pg/mL)	1635 ± 1084	1562 ± 1057	0.18
Number of oocytes retrieved	5.8 ± 4.4	5.9 ± 4.6	0.90
Number of oocytes retrieved/1000 IU of FSH	2.8 ± 4.1	2.2 ± 2.6	0.09
Percentage of mature oocytes	75 (50–100)	71 (50–100)	0.25
Sperm Count (×10^6^/mL)	31.2 ± 28.2	37.5 ± 26.9	<0.001
% Motility (a + b)	35 (20–50)	40 (30–50)	<0.001

Data are presented as mean ± standard deviation or median (interquartile range). IU: International Units; BMI: Body Mass Index; AMH: Anti-Müllerian Hormone; FSH: Follicle-Stimulating Hormone; hCG: Human Chorionic Gonadotropin.

**Table 2 diagnostics-10-00083-t002:** ART outcomes in the two studied groups.

Header Parameters	Controls *n* = 851	Endometriosis *n* = 429	Estimated Difference	95% CI Lower Limit	95% CI Upper Limit	*p*-Value	corrected *p*-Value *
Fertilization rate, median (IQR)	75 (50–100)	75 (50–100)	0.0	−4.5	4.5	0.29	0.85
Cleavage rate, median (IQR)	100 (100–100)	100 (100–100)	0.0	0.0	0.0	0.33	0.83
Clevage stage embryos (*n*), mean ± SD	3.5 ± 2.6	3.7 ± 2.6	0.1	−0.2	0.5	0.42	0.77
Number of cells of the embryos, mean ± SD	7.0 ± 1.4	6.9 ± 1.5	−0.1	-0.3	0.1	0.22	0.42
Percentage of good/fair embryos, median (IQR)	56 (25–100)	50 (17–88)	−5.6	−15.0	4.0	0.20	0.36
Blastulation rate, median (IQR)	50 (25–67)	50 (25–67)	0.0	−6.0	6.0	0.68	0.22
Percentage of good/fair blastocysts, median (IQR)	50 (33–80)	50 (25–75)	0.0	−7.0	7.0	0.43	0.88

Data are presented as mean ± standard deviation or median (IQR: interquartile range). * Adjusted for age, BMI, semen parameters and percentage of mature oocytes.

**Table 3 diagnostics-10-00083-t003:** Embryo transfer details and ongoing pregnancy rate.

Header Parameters	Controls *n* = 516	Endometriosis *n* = 253	*p*-Value
Number of transferred embryos, mean ± SD	1.6 ± 0.7	1.7 ± 0.7	0.10
Day 3 transfers	1.7 ± 0.7	1.8 ± 0.7	0.16
Day 5 transfers	1.3 ± 0.5	1.4 ± 0.5	0.20
Number of transfers (%)			
Day 3 transfers	396 (76.7)	192 (75.9)	0.86
Day 5 transfers	120 (23.3)	61 (24.1)	
Ongoing pregnancy rate (95% CI)			
All transfers	24.2 (20.7–28.1)	17.8 (13.5–22.9)	0.05
Day 3 transfers	21.7 (17.9–26.0)	15.1 (10.7–20.9)	0.07
Day 5 transfers	32.5 (24.8–41.3)	26.2 (16.8–38.4)	0.49
Adjusted Odds Ratios for ongoing pregnancy rate, (95% CI)			
All transfers *	-	0.62 (0.40–0.94)	0.02
Day 3 transfers **	-	0.58 (0.35–0.97)	0.04
Day 5 transfers **	-	0.76 (0.35–1.64)	0.49

* Adjusted for age, BMI, semen parameters, percentage of mature oocytes, day of the transfer and number of transferred embryos. ** Adjusted for age, BMI, semen parameters, percentage of mature oocytes and number of transferred embryos.

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
