# Peer review of "Does Endometriosis Influence the Embryo Quality and/or Development? Insights from a Large Retrospective Matched Cohort Study"

_diagnostics, 2020, doi:10.3390/diagnostics10020083_

Round 1

Reviewer 1 Report

Dear colleagues,

this is an excellent paper which should be published rapidly in the current form.

The study is well designed and presented.

Thank you and best wishes.

Robert Lachmann.

Author Response

We would like to thank the reviewer for the comment.

Reviewer 2 Report

"Does endometriosis influence the embryo quality and/or development? Insights from a large retrospective matched cohort study." presents the result of a single-center historical cohort study that intends to asseess if the quality and the development of embrios achieved by ART is affected by maternal endometriosis.

The study design is correctely presented in the title (following the "American terminology") but not in the main text; this must be corrected.

The study sample (not "study population") must be better described and the options justified. The arguments presented on the discussions over bias control and present bias are insufficient.

The ascertainment of individuals exposed or not to the condition of interest (endometriosis) should be more deeply discussed, as well as the consequences of the options made by the researchers.

Which was the operational definition of endometriosis followed in this study? Does it comply with international recommendations? Which ascertainment of the diagnosis is expected and which was effectively achieved?

Why are only cycles where ICSI was performed selected for the study? How many cycles were excluded due to this selection criteria?

Why are patients (implicitely assumed as the female partner) with diagnosed genetic causes of infertility (described as monogenic) included? Which genetic conditions were identified? How can this affect the outcomes and how to interpret it? Isn´t it a cause of uncontrolled bias? Wouldn't it be more prudent to exclude those few cases?

How do these inclusion criteria affect the representativeness of the study population by the study sample? How does it affect the interpretation of the results? 

Each procedure to which the study sample (persons and cycles) was submitted must be explictely specified wether it followed an established institutional procedure or was left to arbitrary clinical decision.

The effective dimension of the study sample must always be correctely stated whether it refers to persons, to cycles or to products of fecundation and as it changes through the time-lapse covered by the fertilization process. Both in the main text and in the tables, this must be explicit and correct.

The authors state that the Normality of numerical variables was tested using Kolmogoroff-Smirnoff test. Nevertheless, the description of several variables using "mean+/-standard deviation" either clearly reveals or rises strong suspitions on a non-Normal distribution. This must be corrected. If the error is not limited to the description of the variables but translates into the use of unadapted testing, the results might be different and the conclusions of the study may also be different.

In my perspective, it is not adequate to assume individuals as continuous variables, particularly when small numbers are available. For instance, when describing and annalysing the number of embrios or cells, for which the range of numbers is narrow, it does not make sense to use "mean+/-standard deviation" as descriptors ("6.9 cells"??? "3.7 embrios"???) and to test is as a continuous variable.

The study folows hystorically two cohorts of "patients/cycles/fecundation products" for univariate estimation of the different ocurrence of the final (or intermediate) outcome between cohorts, it is correct to use relative risk (risk ratio); there is no justification for the use of odds ratio. Odds ratio should be used for multivariate estimation only.

If only cycles where ICSI was performed were selected for the study, what is the rationale for taking into account "concentration and motility of sperm" as independent variables in the multivariable mixed model used to identify factor that could affect the final outcomes?

Focusing on formal issues: there are achoryms and abbreviations that aare not presented; and they must always be presented in the tables. The tables require more accurate and complete titles and legends.

This suggestions are feasible to comply with and would make the manuscript more clinically and scientifically robust.

Author Response

"Does endometriosis influence the embryo quality and/or development? Insights from a large retrospective matched cohort study." presents the result of a single-center historical cohort study that intends to asseess if the quality and the development of embrios achieved by ART is affected by maternal endometriosis.

1) The study design is correctely presented in the title (following the "American terminology") but not in the main text; this must be corrected.

We would like to thank the reviewer for the comment. We have added information and corrected the main text.

2) The study sample (not "study population") must be better described and the options justified. The arguments presented on the discussions over bias control and present bias are insufficient.

We would like to thank the reviewer for the comment. We have better described the study sample in the text.  The discussion regarding the biases of the study has been made more comprehensive.

3) The ascertainment of individuals exposed or not to the condition of interest (endometriosis) should be more deeply discussed, as well as the consequences of the options made by the researchers.

We thank the reviewer for the comment.  A comment has been added in the discussion.

4) Which was the operational definition of endometriosis followed in this study? Does it comply with international recommendations? Which ascertainment of the diagnosis is expected and which was effectively achieved?

The operational definition of endometriosis followed the classification proposed for the American Society for Reproductive Medicine (ASRM). However, one of the negative point of the ASRM classification is that the disease severity according this classification does not correlate with the location of the disease and the severity of the symptoms. For these reasons, there is currently an increase criticism of this classification. Moreover, the classification does not permit to use it as a tool for a predictive prognosis (Johnson et al., 2017). In addition, this classification system fails to report any extra pelvic endometriosis (Zondervan et al., 2018). Nonetheless, this is the best-known international classification for the disease.

Endometriosis patients that performed ART cycles in our Center were previously diagnosed with endometriosis. In our study, all women received surgical treatment for endometriosis and disease was surgically removed.

5) Why are only cycles where ICSI was performed selected for the study? How many cycles were excluded due to this selection criteria?

As described in the Materials and Methods section, only ICSI cycles were included in the study in order to avoid biases in the evaluation of the number of MII oocytes and in the fertilization rate. In our Unit we perform ICSI for the majority of the cycles. We perform only 9% of conventional FIV cycles. The percentage of FIV/ICSI that we perform is not different between endometriosis patients and not endometriosis patients. The percentage of excluded cycles has been added in the text.

6) Why are patients (implicitely assumed as the female partner) with diagnosed genetic causes of infertility (described as monogenic) included? Which genetic conditions were identified? How can this affect the outcomes and how to interpret it? Isn´t it a cause of uncontrolled bias? Wouldn't it be more prudent to exclude those few cases?

Unfortunately this concept was not clear and the text has been modified to make it clearer. The women referred to by the indication "genetics" in supplementary table 3 were patients who had performed PGT-M cycles for monogenic diseases. The included patients have genetic diseases that do not affect the fertility of the couple.

7) How do these inclusion criteria affect the representativeness of the study population by the study sample? How does it affect the interpretation of the results?

We thank the reviewer for the comment. The discussion was improved by adding comments on the presented topics.

8) Each procedure to which the study sample (persons and cycles) was submitted must be explictely specified wether it followed an established institutional procedure or was left to arbitrary clinical decision.

We thank the reviewer for the comment.  All the procedures were carried out as reported in the text with the relative literature. We also described the criteria based on which the decision to extend the embryo culture until day 5 was established. Briefly, the criteria for blastocyst culture were the presence at day 3 of 4 or more embryos and at least 2 good/fair embryos in women younger than 38 years old and three good/fair embryos for women aged 38 years and older. This decision was based on the scoring system described in the Istanbul consensus workshop on embryo assessment [33] and following the guidelines for practice reported by the British Fertility Society and Association of Clinical Embryologists [34].

9) The effective dimension of the study sample must always be correctely stated whether it refers to persons, to cycles or to products of fecundation and as it changes through the time-lapse covered by the fertilization process. Both in the main text and in the tables, this must be explicit and correct.

We thank the reviewer for the comment. We have made changes to the text so that it is now clear what the data refer to.

10) The authors state that the Normality of numerical variables was tested using Kolmogoroff-Smirnoff test. Nevertheless, the description of several variables using "mean+/-standard deviation" either clearly reveals or rises strong suspitions on a non-Normal distribution. This must be corrected. If the error is not limited to the description of the variables but translates into the use of unadapted testing, the results might be different and the conclusions of the study may also be different.

We have described all the continuous variables except percentage as mean ± standard deviation. This is not an error, and is indeed very common in this field (eg. Wei et al., Lancet, 2019),  even for those variables with a distribution that deviates slightly from normality. This was done to make the data more readable and comparable.  However, the p values reported were obtained by applying the appropriate test according to the distribution, as described in the materials and methods. Multivariate analysis was conducted using the generalized estimating equation (GEE) approach, thus making possible to analyze variables with non-normal distribution.

11) In my perspective, it is not adequate to assume individuals as continuous variables, particularly when small numbers are available. For instance, when describing and annalysing the number of embrios or cells, for which the range of numbers is narrow, it does not make sense to use "mean+/-standard deviation" as descriptors ("6.9 cells"??? "3.7 embrios"???) and to test is as a continuous variable.

As previously mentioned, only the description of the data uses a form usually associated with the normal distribution. It is common, for example, to describe the average number of embryos transferred for a certain group of patients as the mean ± standard deviation (eg  1.05 ± 0.22 embryos, Wei et al., Lancet, 2019). Both the number of cells and the number of embryos are continuous numbers, the distribution of which is however not perfectly normal. The use of GEEs allowed us to analyze the data in the proper way.

12) The study folows hystorically two cohorts of "patients/cycles/fecundation products" for univariate estimation of the different ocurrence of the final (or intermediate) outcome between cohorts, it is correct to use relative risk (risk ratio); there is no justification for the use of odds ratio. Odds ratio should be used for multivariate estimation only.

All the odds ratios reported in the paper  refer to results of a multivariate analysis. Both the “materials and methods” section and the individual tables report the factors for which the data has been corrected.

13) If only cycles where ICSI was performed were selected for the study, what is the rationale for taking into account "concentration and motility of sperm" as independent variables in the multivariable mixed model used to identify factor that could affect the final outcomes?

We would like to thank the reviewer for the comment. So far, little is known about the impact of abnormal sperm parameters on the developmental competence of the embryo. However some studies have reported a negative impact of motility on the fertilization rate and a significantly reduced blastocyst formation rate for semen concentration <1M/ml (Borges E J et al., 2016 ; Bartolacci et al., 2018).

In table 1 we reported a significant difference between the two study samples regarding sperm count and motility. We therefore considered it appropriate to correct the results for these covariates.

14) Focusing on formal issues: there are achoryms and abbreviations that aare not presented; and they must always be presented in the tables. The tables require more accurate and complete titles and legends.

We thank the reviewer for the comment. We have added the missing information in the tables.

This suggestions are feasible to comply with and would make the manuscript more clinically and scientifically robust.

Reviewer 3 Report

I reviewed the manuscript “Does endometriosis influence the embryo quality and/or development? Insights from a large retrospective matched cohort study" (diagnostics-697442). The objective of the manuscript was to evaluate the potential effect of endometriosis in the development of embryos at cleave stage in assisted reproduction treatment cycles., according to the authors. The research was approved by the Local Institutional Review Board and a total of 309 women were included in the study with a total of 429 ART cycles performed. The statistical analysis is adequate, and the results have been adequately described. I think this paper is suitable for publication in its current form.

Author Response

(The authors gave the same response as above.)
